# Motion Transformer with Global Intention Localization and Local Movement Refinement

**Shaoshuai Shi,  Li Jiang,  Dengxin Dai,  Bernt Schiele**
Max Planck Institute for Informatics, Saarland Informatics Campus
{sshi, lijiang, ddai, schiele}@mpi-inf.mpg.de

## Abstract

Predicting multimodal future behavior of traffic participants is essential for robotic vehicles to make safe decisions. Existing works explore to directly predict future trajectories based on latent features or utilize dense goal candidates to identify agent's destinations, where the former strategy converges slowly since all motion modes are derived from the same feature while the latter strategy has efficiency issue since its performance highly relies on the density of goal candidates. In this paper, we propose the Motion TRansformer (MTR) framework that models motion prediction as the joint optimization of global intention localization and local movement refinement. Instead of using goal candidates, MTR incorporates spatial intention priors by adopting a small set of learnable motion query pairs. Each motion query pair takes charge of trajectory prediction and refinement for a specific motion mode, which stabilizes the training process and facilitates better multimodal predictions. Experiments show that MTR achieves state-of-the-art performance on both the marginal and joint motion prediction challenges, ranking $1^{st}$ on the leaderboards of Waymo Open Motion Dataset. Code will be available at https://github.com/sshaoshuai/MTR.

## 1   Introduction

Motion forecasting is a fundamental task of modern autonomous driving systems. It has been receiving increasing attention in recent years [19, 46, 29, 57, 35] as it is crucial for robotic vehicles to understand driving scenes and make safe decisions. Motion forecasting requires to predict future behaviors of traffic participants by jointly considering the observed agent states and road maps, which is challenging due to inherently multimodal behaviors of the agent and complex scene environments.

To cover all potential future behaviors of the agent, existing approaches mainly fall into two different lines: the goal-based methods and the direct-regression methods. The goal-based methods [19, 61] adopt dense goal candidates to cover all possible destinations of the agent, predicting the probability of each candidate being a real destination and then completing the full trajectory for each selected candidate. Although these goal candidates alleviate the burden of model optimization by reducing trajectory uncertainty, their density largely affects the performance of these methods: fewer candidates will decrease the performance while more candidates will greatly increase computation and memory cost. Instead of using goal candidates, the direct-regression methods [35, 47] directly predict a set of trajectories based on the encoded agent feature, covering the agent's future behavior adaptively. Despite the flexibility in predicting a broad range of agent behaviors, they generally converge slowly as various motion modes are required to be regressed from the same agent feature without utilizing any spatial priors. They also tend to predict the most frequent modes of training data since these frequent modes dominate the optimization of the agent feature. In this paper, we present a unified framework, namely **Motion TRansformer (MTR)**, which takes the best of both types of methods.

36th Conference on Neural Information Processing Systems (NeurIPS 2022).

In our proposed MTR, we adopt a small set of novel motion query pairs to model motion prediction as the joint optimization of two tasks: The first global intention localization task aims to roughly identify agent's intention for achieving higher efficiency, while the second local movement refinement task aims to adaptively refine each intention's predicted trajectory for achieving better accuracy. Our approach not only stabilizes the training process without depending on dense goal candidates but also enables flexible and adaptive prediction by enabling local refinement for each motion mode.

Specifically, each motion query pair consists of two components, *i.e.*, a static intention query and a dynamic searching query. The static intention queries are introduced for global intention localization, where we formulate them based on a small set of spatially distributed intention points. Each static intention query is the learnable positional embedding of an intention point for generating trajectory of a specific motion mode, which not only stabilizes the training process by explicitly utilizing different queries for different modes, but also eliminates the dependency on dense goal candidates by requiring each query to take charge of a large region. The dynamic searching queries are utilized for local movement refinement, where they are also initialized as the learnable embeddings of the intention points but are responsible for retrieving fine-grained local features around each intention point. For this purpose, the dynamic searching queries are dynamically updated according to the predicted trajectories, which can adaptively gather latest trajectory features from a deformable local region for iterative motion refinement. These two queries complement each other and have been empirically demonstrated their great effectiveness in predicting multimodal future motion. Besides that, we also propose a dense future prediction module. Existing works generally focus on modeling the agent interaction over past trajectories while ignoring the future trajectories' interaction. To compensate for such information, we adopt a simple auxiliary regression head to densely predict future trajectory and velocity for each agent, which are encoded as additional future context features to benefit future motion prediction of our interested agent. The experiments show that this simple auxiliary task works well and remarkably improves the performance of multimodal motion prediction.

Our contributions are three-fold: (1) We propose a novel motion decoder network with a new concept of motion query pair, which adopts two types of queries to model motion prediction as joint optimization of global intention localization and local movement refinement. It not only stabilizes the training with mode-specific motion query pairs, but also enables adaptive motion refinement by iteratively gathering fine-grained trajectory features. (2) We present an auxiliary dense future prediction task to enable the future interactions between our interested agent and other agents. It facilitates our framework to predict more scene-compliant trajectories for the interacting agents. (3) By adopting these techniques, we propose MTR framework that explores transformer encoder-decoder structure for multimodal motion prediction. Our approach achieves state-of-the-art performance on both the marginal and joint motion prediction benchmarks of Waymo Open Motion Dataset (WOMD) [14], outperforming previous best ensemble-free approaches with $+8.48\%$ mAP gains for marginal motion prediction and $+7.98\%$ mAP gains for joint motion prediction. As of 19 May 2022, our approach ranks $1^{st}$ on both the marginal and joint motion prediction leaderboards of WOMD.

## 2   Related Work

**Motion Prediction for Autonomous Driving.** Recently, motion prediction has been extensively studied due to the growing interest in autonomous driving, and it typically takes road map and agent history states as input. To encode such scene context, early works [36, 31, 5, 12, 60, 3, 8] typically rasterize them into an image so as to be processed with convolutional neural networks (CNNs). LaneGCN [27] builds a lane graph toscalability capture map topology. VectorNet [16] is widely adopted by recent works [19, 43, 35, 47] due to its efficiency and scalability, where both road maps and agent trajectories are represented as polylines. We also adopt this vector representation, but instead of building global graph of polylines, we propose to adopt transformer encoder on local connected graph, which not only better maintains input locality structure but also is more memory-efficient to enable larger map encoding for long-term motion prediction.

Given the encoded scene context features, existing works explore various strategies to model multi-modal future motion. Early works [1, 20, 39, 44, 40] propose to generate a set of trajectory samples to approximate the output distribution. Some other works [9, 21, 33, 37, 41] parameterize multi-modal predictions with Gaussian Mixture Models (GMMs) to generate compact distribution. HOME series [18, 17] generate trajectories with sampling on a predicted heatmap. IntentNet [7] considers intention prediction as a classification with 8 high level actions, while [29] proposes a region-based

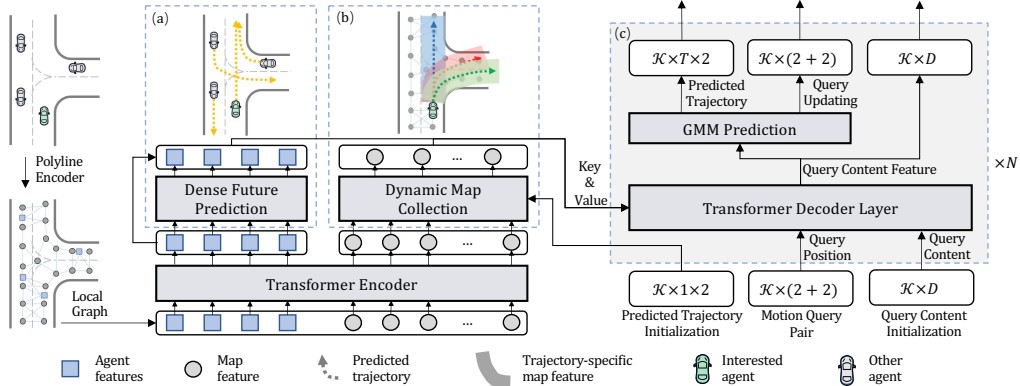

Figure 1: The architecture of MTR framework. (a) indicates the dense future prediction module, which predicts a single trajectory for each agent (*e.g.*, drawn as yellow dashed curves in the above of (a)). (b) indicates the dynamic map collection module, which collects map elements along each predicted trajectory (*e.g.*, drawn as the shadow region along each trajectory in the above part of (b)) to provide trajectory-specific feature for motion decoder network. (c) indicates the motion decoder network, where $\mathcal{K}$ is the number of motion query pairs, $T$ is the number of future frames, $D$ is hidden feature dimension and $N$ is the number of transformer decoder layers. The predicted trajectories, motion query pairs, and query content features are the outputs from last decoder layer and will be taken as input to next decoder layer. For the first decoder layer, both two components of motion query pair are initialized as predefined intention points, the predicted trajectories are replaced with the intention points for initial map collection, and query content features are initialized as zeros.

training strategy. Goal-based methods [61, 40, 15, 30] are another kinds of models where they first estimate several goal points of the agents and then complete full trajectory for each goal.

Recently, the large-scale Waymo Open Motion Dataset (WOMD) [14] is proposed for long-term motion prediction. To address this challenge, DenseTNT [19] adopts a goal-based strategy to classify endpoint of trajectory from dense goal points. Other works directly predict the future trajectories based on the encoded agent features [35] or latent anchor embedding [47]. However, the goal-based strategy has the efficiency concern due to a large number of goal candidates, while the direct-regression strategy converges slowly as the predictions of various motion modes are regressed from the same agent feature. In contrast, our approach adopts a small set of learnable motion query pairs, which not only eliminate the large number of goal candidates but also alleviate the optimization burden by utilizing mode-specific motion query pairs for predicting different motion modes.

Some very recent works [45, 23, 22] also achieve top performance on WOMD by exploring Mix-and-Match block [45], a variant of MultiPath++ [23] or heterogeneous graph [22]. However, they generally focus on exploring various structures for encoding scene context, while how to design a better motion decoder for multimodal motion prediction is still underexplored. In contrast, our approach focuses on addressing this challenge with a novel transformer-based motion decoder network.

**Transformer.** Transformer [48] has been widely applied in natural language processing [11, 2] and computer vision [13, 50, 4, 49, 58]. Our approach is inspired by DETR [4] and its follow-up works [63, 32, 56, 25, 28, 10, 59], especially DAB-DETR [28], where the object query is considered as the positional embedding of a spatial anchor box. Motivated by their great success in object detection, we introduce a novel concept of motion query pair to model multimodal motion prediction with prior intention points, where each motion query pair takes charge of predicting a specific motion mode and also enables iterative motion refinement by combining with transformer decoders.

## 3 Motion TRansformer (MTR)

We propose Motion TRansformer (MTR), which adopts a novel transformer encoder-decoder structure with iterative motion refinement for predicting multimodal future motion. The overall structure is illustrated in Figure 1. In Sec. 3.1, we introduce our encoder network for scene context modeling. In Sec. 3.2, we present motion decoder network with a novel concept of motion query pair for predicting multimodal trajectories. Finally, in Sec. 3.3, we introduce the optimization process of our framework.

## 3.1 Transformer Encoder for Scene Context Modeling

The future behaviors of the agents highly depend on the agents' interaction and road map. To encode such scene context, existing approaches have explored various strategies by building global interacting graph [16, 19] or summarizing map features to agent-wise features [35, 47]. We argue that the locality structure is important for encoding scene context, especially for the road map. Hence, we propose a transformer encoder network with local self-attention to better maintain such structure information.

**Input representation.** We follow the vectorized representation [16] to organize both input trajectories and road map as polylines. For the motion prediction of a interested agent, we adopt the agent-centric strategy [61, 19, 47] that normalizes all inputs to the coordinate system centered at this agent. Then, a simple polyline encoder is adopted to encode each polyline as an input token feature for the transformer encoder. Specifically, we denote the history state of $N_a$ agents as $A_{\text{in}} \in \mathbb{N}^{N_a \times t \times C_a}$, where $t$ is the number of history frames, $C_a$ is the number of state information (*e.g.*, location, heading angle and velocity), and we pad zeros at the positions of missing frames for trajectories that have less than $t$ frames. The road map is denoted as $M_{\text{in}} \in \mathbb{R}^{N_m \times n \times C_m}$, where $N_m$ is the number of map polylines, $n$ is the number of points in each polyline and $C_m$ is the number of attributes of each point (*e.g.*, location and road type). Both of them are encoded by a PointNet-like [38] polyline encoder as:

$$A_{\text{p}} = \phi\left(\text{MLP}(A_{\text{in}})\right), \quad M_{\text{p}} = \phi\left(\text{MLP}(M_{\text{in}})\right), \tag{1}$$

where $\text{MLP}(\cdot)$ is a multilayer perceptron network, and $\phi$ is max-pooling to summarize each polyline features as agent features $A_{\text{p}} \in \mathbb{R}^{N_a \times D}$ and map features $M_{\text{p}} \in \mathbb{R}^{N_m \times D}$ with feature dimension $D$.

**Scene context encoding with local transformer encoder.** The local structure of scene context is important for motion prediction. For example, the relation of two parallel lanes is important for modelling the motion of changing lanes, but adopting attention on global connected graph equally considers relation of all lanes. In contrast, we introduce such prior knowledge to context encoder by adopting local attention, which better maintains the locality structure and are more memory-efficient. Specifically, the attention module of $j$-th transformer encoder layer can be formulated as:

$$G^j = \text{MultiHeadAttn}\left(\text{query}=G^{j-1} + \text{PE}_{G^{j-1}}, \text{ key}=\kappa(G^{j-1}) + \text{PE}_{\kappa(G^{j-1})}, \text{ value}=\kappa(G^{j-1})\right), \tag{2}$$

where $\text{MultiHeadAttn}(\cdot, \cdot, \cdot)$ is the multi-head attention layer [48], $G^0 = [A_{\text{p}}, M_{\text{p}}] \in \mathbb{N}^{(N_a + N_m) \times D}$ concatenating the features of agents and map, and $\kappa(\cdot)$ denotes $k$-nearest neighbor algorithm to find $k$ closest polylines for each query polyline. PE denotes sinusoidal position encoding of input tokens, where we utilize the latest position for each agent and utilize polyline center for each map polyline. Thanks to such local self-attention, our framework can encode a much larger area of scene context.

The encoder network finally generates both agent features $A_{\text{past}} \in \mathbb{R}^{N_a \times D}$ and map features $M \in \mathbb{R}^{N_m \times D}$, which are considered as the scene context inputs of the following decoder network.

**Dense future prediction for future interactions.** Interactions with other agents heavily affect behaviors of our interested agent, and previous works propose to model the multi-agent interactions with hub-host based network [64], dynamic relational reasoning [26], social spatial-temporal network [55], etc. However, most existing works generally focus on learning such interactions over past trajectories while ignoring the interactions of future trajectories. Therefore, considering that the encoded features $A$ have already learned rich context information of all agents, we propose to densely predict both future trajectories and velocities of all agents by adopting a simple regression head on $A$:

$$S_{1:T} = \text{MLP}(A_{\text{past}}), \tag{3}$$

where $S_i \in \mathbb{R}^{N_a \times 4}$ includes future position and velocity of each agent at time step $i$, and $T$ is the number of future frames to be predicted. The predicted trajectories $S_{1:T}$ are encoded by adopting the same polyline encoder as Eq. (1) to encode the agents' future states as features $A_{\text{future}} \in \mathbb{R}^{N_a \times D}$, which are then utilized to enhance the above features $A$ by using a feature concatenation and three MLP layers as $A = \text{MLP}([A_{\text{past}}, A_{\text{future}}])$. This auxiliary task provides additional future context information to the decoder network, facilitating the model to predict more scene-compliant future trajectories for the interested agent. The experiments in Table 3 demonstrates that this simple and light-weight auxiliary task can effectively improve the performance of multimodal motion prediction.

## 3.2 Transformer Decoder with Motion Query Pair

Given the scene context features, a transformer-based motion decoder network is adopted for multi-modal motion prediction, where we propose *motion query pair* to model motion prediction as the

joint optimization of global intention localization and local movement refinement. Each motion query pair contains two types of queries, *i.e.*, static intention query and dynamic searching query, for conducting global intention localization and local movement refinement respectively. As shown in Figure 2, our motion decoder network contains stacked transformer decoder layers for iteratively refining the predicted trajectories with motion query pairs. Next, we illustrate the detailed structure.

**Global intention localization** aims to localize agent's potential motion intentions in an efficient and effective manner. We propose *static intention query* to narrow down the uncertainty of future trajectory by utilizing different intention queries for different motion modes. Specifically, we generate $\mathcal{K}$ representative intention points $I \in \mathbb{R}^{\mathcal{K} \times 2}$ by adopting k-means clustering algorithm on the endpoints of ground-truth (GT) trajectories, where each intention point represents an implicit motion mode that considers both motion direction and velocity. We model each static intention query as the learnable positional embedding of the intention point as:

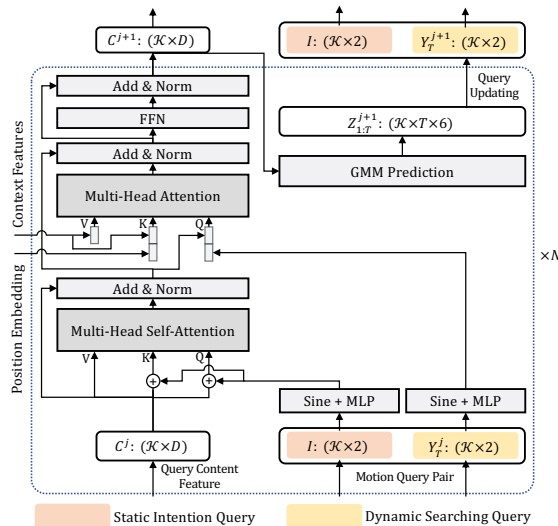

$$Q_I = \text{MLP}\left(\text{PE}(I)\right), \qquad (4)$$

where $\text{PE}(\cdot)$ is the sinusoidal position encoding, and $Q_I \in \mathbb{R}^{\mathcal{K} \times D}$. Notably, each intention query takes charge of predicting trajectories for a specific motion mode, which stabilizes the training process and facilitates predicting multimodal trajectories since each motion mode has their own learnable embedding. Thanks to their learnable and adaptive properties, we only need a small number of queries (*e.g.*, 64 queries in our setting) for efficient intention localization, instead of using densely-placed goal candidates [61, 19] to cover the destinations of the agents.

Figure 2: The network structure of our motion decoder network with motion query pair.

**Local movement refinement** aims to complement with global intention localization by iteratively gathering fine-grained trajectory features for refining the trajectories. We propose *dynamic searching query* to adaptively probe trajectory features for each motion mode. Each dynamic searching query is also the position embedding of a spatial point, which is initialized with its corresponding intention point but will be dynamically updated according to the predicted trajectory in each decoder layer. Specifically, given the predicted future trajectories $Y_{1:T}^j = \{Y_i^j \in \mathbb{R}^{\mathcal{K} \times 2} \mid i = 1, \cdots, T\}$ in $j$-th decoder layer, the dynamic searching query of $(j + 1)$-th decoder layer is updated as follows:

$$Q_S^{j+1} = \text{MLP}\left(\text{PE}(Y_T^j)\right). \qquad (5)$$

As shown in Figure 3, for each motion query pair, we propose a *dynamic map collection* module to extract fine-grained trajectory features by querying map features from a trajectory-aligned local region, which is implemented by collecting $L$ polylines whose centers are closest to the predicted trajectory. As the agent's behavior largely depends on road maps, this local movement refinement strategy enables to continually focus on latest local context information for iterative motion refinement.

**Attention module with motion query pair.** In each decoder layer, static intention query is utilized to propagate information among different motion intentions, while dynamic searching query is utilized to aggregate trajectory-specific features from scene context features. Specifically, we utilize static intention query as the position embedding of self-attention module as follows:

$$C_{\text{sa}}^j = \text{MultiHeadAttn}(\text{query}=C^{j-1} + Q_I, \text{ key}=C^{j-1} + Q_I, \text{ value}=Q_I), \qquad (6)$$

where $C^{j-1} \in \mathbb{R}^{\mathcal{K} \times D}$ is query content features from $(j-1)$-th decoder layer, $C^0$ is initialized to zeros, and $C_{\text{sa}}^j \in \mathbb{R}^{\mathcal{K} \times D}$ is the updated query content. Next, we utilize dynamic searching query as query position embedding of cross attention to probe trajectory-specific features from the outputs of encoder. Inspired by [32, 28], we concatenate content features and position embedding for both query and key to decouple their contributions to the attention weights. Two cross-attention modules

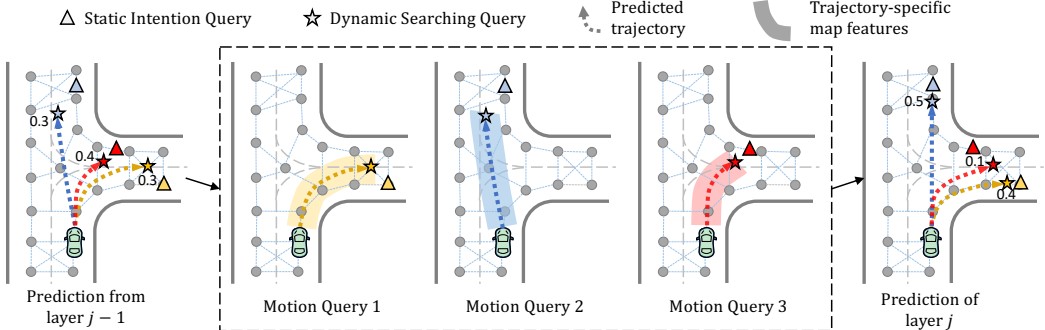

Figure 3: The illustration of dynamic map collection module for iterative motion refinement.

are adopted separately for aggregating features from both agent features $A$ and map features $M$ as:

$$C_A^j = \text{MultiHeadAttn}(\text{query}=[C_{\text{sa}}^j, Q_S^j], \text{ key}=[A, \text{PE}_A], \text{ value}=A),$$

$$C_M^j = \text{MultiHeadAttn}(\text{query}=[C_{\text{sa}}^j, Q_S^j], \text{ key}=[\alpha(M), \text{PE}_{\alpha(M)}], \text{ value}=\alpha(M)), \tag{7}$$

$$C^j = \text{MLP}([C_A^j, C_M^j])$$

where $[\cdot, \cdot]$ indicates feature concatenation, $\alpha(M)$ is the aforementioned dynamic map collection module to collect $L$ trajectory-aligned map features for motion refinement. Note that for simplicity, in Eq. (6) and (7), we omit the residual connection and feed-forward network in transformer layer [48].

Finally, $C^j \in \mathbb{R}^{\mathcal{K} \times D}$ is the updated query content features for each motion query pair in $j$-th layer.

**Multimodal motion prediction with Gaussian Mixture Model.** For each decoder layer, we append a prediction head to $C^j$ for generating future trajectories. As the behaviors of the agents are highly multimodal, we follow [9, 47] to represent the distribution of predicted trajectories with Gaussian Mixture Model (GMM) at each time step. Specifically, for each future time step $i \in \{1, \cdots, T\}$, we predict the probability $p$ and parameters $(\mu_x, \mu_y, \sigma_x, \sigma_y, \rho)$ of each Gaussian component as follows

$$Z_{1:T}^j = \text{MLP}(C^j), \tag{8}$$

where $Z_i^j \in \mathbb{R}^{\mathcal{K} \times 6}$ includes $\mathcal{K}$ Gaussian components $\mathcal{N}_{1:\mathcal{K}}(\mu_x, \sigma_x; \mu_y, \sigma_y; \rho)$ with probability distribution $p_{1:\mathcal{K}}$. The predicted distribution of agent's position at time step $i$ can be formulated as:

$$P_i^j(o) = \sum_{k=1}^{\mathcal{K}} p_k \cdot \mathcal{N}_k(o_x - \mu_x, \sigma_x; o_y - \mu_y, \sigma_y; \rho). \tag{9}$$

where $P_i^j(o)$ is the occurrence probability of the agent at spatial position $o \in \mathbb{R}^2$. The predicted trajectories $Y_{1:T}^j$ can be generated by simply extracting the predicted centers of Gaussian components.

### 3.3 Training Losses

Our model is trained end-to-end with two training losses. The first auxiliary loss is $L1$ regression loss to optimize the outputs of Eq. (3). For the second Gaussian regression loss, we adopt negative log-likelihood loss according to Eq. (9) to maximum the likelihood of ground-truth trajectory. Inspired by [9, 47], we adopt a hard-assignment strategy that selects one closest motion query pair as positive Gaussian component for optimization, where the selection is implemented by calculating the distance between each intention point and the endpoint of GT trajectory. The Gaussian regression loss is adopted in each decoder layer, and the final loss is the sum of the auxiliary regression loss and all the Gaussian regression loss with equal loss weights. Please refer to appendix for more loss details.

## 4 Experiments

### 4.1 Experimental Setup

**Dataset and metrics.** We evaluate our approach on the large-scale Waymo Open Motion Dataset (WOMD) [14], which mines interesting interactions from real-world traffic scenes and is currently

Table 1: Performance comparison of marginal motion prediction on the validation and test set of Waymo Open Motion Dataset. †: The results are shown in *italic* for reference since their performance is achieved with model ensemble techniques. We only evaluate our default setting MTR on the test set by submitting to official test server due to the limitation of submission times of WOMD.

| | Method | Reference | minADE ↓ | minFDE ↓ | Miss Rate ↓ | mAP ↑ |
|---|---|---|---|---|---|---|
| Test | MotionCNN [24] | CVPRw 2021 | 0.7400 | 1.4936 | 0.2091 | 0.2136 |
| | ReCoAt [62] | CVPRw 2021 | 0.7703 | 1.6668 | 0.2437 | 0.2711 |
| | DenseTNT [19] | ICCV 2021 | 1.0387 | 1.5514 | 0.1573 | 0.3281 |
| | SceneTransformer [35] | ICLR 2022 | 0.6117 | **1.2116** | 0.1564 | 0.2788 |
| | MTR (Ours) | - | **0.6050** | 1.2207 | **0.1351** | **0.4129** |
| | †MultiPath++ [47] | ICRA 2022 | *0.5557* | *1.1577* | *0.1340* | *0.4092* |
| | †MTR-Advanced-ens (Ours) | - | *0.5640* | *1.1344* | *0.1160* | *0.4492* |
| Val | MTR (Ours) | - | 0.6046 | 1.2251 | 0.1366 | **0.4164** |
| | MTR-e2e (Ours) | - | **0.5160** | **1.0404** | **0.1234** | 0.3245 |
| | †MTR-ens (Ours) | - | *0.5686* | *1.1534* | *0.1240* | *0.4323* |
| | †MTR-Advanced-ens (Ours) | - | *0.5597* | *1.1299* | *0.1167* | *0.4551* |

Table 2: Performance comparison of joint motion prediction on the interactive validation and test set of Waymo Open Motion Dataset.

| | Method | Reference | minADE ↓ | minFDE ↓ | Miss Rate ↓ | mAP ↑ |
|---|---|---|---|---|---|---|
| Test | Waymo LSTM baseline [14] | ICCV 2021 | 1.9056 | 5.0278 | 0.7750 | 0.0524 |
| | HeatIRm4 [34] | CVPRw 2021 | 1.4197 | 3.2595 | 0.7224 | 0.0844 |
| | AIR$^2$ [54] | CVPRw 2021 | 1.3165 | 2.7138 | 0.6230 | 0.0963 |
| | SceneTransformer [35] | ICLR 2022 | 0.9774 | 2.1892 | 0.4942 | 0.1192 |
| | M2I [43] | CVPR 2022 | 1.3506 | 2.8325 | 0.5538 | 0.1239 |
| | MTR (Ours) | - | **0.9181** | **2.0633** | **0.4411** | **0.2037** |
| Val | MTR (Ours) | - | 0.9132 | 2.0536 | 0.4372 | 0.1992 |

the most diverse interactive motion dataset. There are two tasks in WOMD with separate evaluation metrics: (1) The *marginal motion prediction challenge* that independently evaluates the predicted motion of each agent (up to 8 agents per scene). (2) The *joint motion prediction challenge* that needs to predict the joint future positions of 2 interacting agents for evaluation. Both of them provide 1 second of history data and aim to predict 6 marginal or joint trajectories of the agents for 8 seconds into the future. There are totally $487k$ training scenes, and about $44k$ validation scenes and $44k$ testing scenes for each challenge. We utilize the official evaluation tool to calculate the evaluation metrics, where the mAP and miss rate are the most important ones as in the official leaderboard[52, 51].

**Implementation details.** For the context encoding, we stack 6 transformer encoder layers. The road map is represented as multiple polylines, where each polyline contains up to 20 points (about $10m$ in WOMD). We select $N_m = 768$ nearest map polylines around the interested agent. The number of neighbors in encoder's local self-attention is set to 16. The encoder hidden feature dimension is set as $D = 256$. For the decoder modules, we stack 6 decoder layers. $L$ is set to 128 to collect the closest map polylines from context encoder for motion refinement. By default, we utilize 64 motion query pairs where their intention points are generated by conducting k-means clustering algorithm on the training set. To generate 6 future trajectories for evaluation, we use non-maximum suppression (NMS) to select top 6 predictions from 64 predicted trajectories by calculating the distances between their endpoints, and the distance threshold is set as $2.5m$. Please refer to Appendix for more details.

**Training details.** Our model is trained in an end-to-end manner by AdamW optimizer with a learning rate of 0.0001 and batch size of 80 scenes. We train the model for 30 epochs with 8 GPUs (NVIDIA RTX 8000), and the learning rate is decayed by a factor of 0.5 every 2 epochs from epoch 20. The weight decay is set as 0.01 and we do not use any data augmentation.

**MTR-e2e for end-to-end motion prediction.** We also propose an end-to-end variant of MTR, called MTR-e2e, where only 6 motion query pairs are adopted so as to remove NMS post processing. In the training process, instead of using static intention points for target assignment as in MTR, MTR-e2e selects positive mixture component by calculating the distances between its 6 predicted trajectories and the GT trajectory, since 6 intention points are too sparse to well cover all potential future motions.

## 4.2 Main Results

**Performance comparison for marginal motion prediction.** Table 1 shows our main results for marginal motion prediction, our MTR outperforms previous ensemble-free approaches [19, 35] with

Table 3: Effects of different components in MTR framework. All models share the same encoder network. "latent learnable embedding" indicates using 6 latent learnable embeddings as queries of decoder network, and "iterative refinement" indicates using 6 stacked decoders for motion refinement.

| Global Intention Localization | Iterative Refinement | Local Movement Refinement | Dense Future Prediction | minADE ↓ | minFDE ↓ | Miss Rate ↓ | **mAP** ↑ |
|---|---|---|---|---|---|---|---|
| Latent learnable embedding | × | × | × | 0.6829 | 1.4841 | 0.2128 | 0.2633 |
| Static intention query | × | × | × | 0.7036 | 1.4651 | 0.1845 | 0.3059 |
| Static intention query | ✓ | × | × | 0.6919 | 1.4217 | 0.1776 | 0.3171 |
| Static intention query | ✓ | ✓ | × | 0.6833 | 1.4059 | 0.1756 | 0.3234 |
| Static intention query | ✓ | × | ✓ | 0.6735 | 1.3847 | 0.1706 | 0.3284 |
| Static intention query | ✓ | ✓ | ✓ | **0.6697** | **1.3712** | **0.1668** | **0.3437** |

remarkable margins, increasing the mAP by $+8.48\%$ and decreasing the miss rate from $15.64\%$ to $13.51\%$. In particular, our single-model results of MTR also achieve better mAP than the latest work MultiPath++ [47], where it uses a novel model ensemble strategy that boosts its performance.

Table 1 also shows the comparison of MTR variants. MTR-e2e achieves better minADE and minFDE by removing NMS post-processing, while MTR achieves better mAP since it learns explicit meaning of each motion query pair that produces more confident intention predictions. We also propose a simple model ensemble strategy to merge the predictions of MTR and MTR-e2e and utilize NMS to remove redundant predictions (denoted as MTR-ens), and it takes the best of both models and achieves much better mAP. By adopting such ensemble strategy to 7 variants of our framework (*e.g.*, more decoder layers, different number of queries, larger hidden dimension), our advanced ensemble results (denoted as MTR-Advanced-ens) achieve best performance on the test set leaderboard.

**Performance comparison for joint motion prediction.** To evaluate our approach for joint motion prediction, we combine the marginal predictions of two interacting agents into joint prediction as in [6, 14, 43], where we take the top 6 joint predictions from 36 combinations of these two agents. The confidence of each combination is the product of marginal probabilities. Table 2 shows that our approach outperforms state-of-the-arts [35, 43] with large margins on all metrics. Particularly, our MTR boosts the mAP from $12.39\%$ to $20.37\%$ and decreases the miss rate from $49.42\%$ to $44.11\%$. The remarkable performance gains demonstrate the effectiveness of MTR for predicting scene-consistent future trajectories. Besides that, we also provide some qualitative results in Figure 5 to show our predictions in complicated interacting scenarios.

As of May 19, 2022, our MTR ranks $1^{st}$ on the motion prediction leaderboard of WOMD for both two challenges [52, 51]. Our approach with more ensembled variants of MTR (*i.e.*, MTR-Advacned-ens) also won the champion of Motion Prediction Challenge in Waymo Open Dataset Challenge 2022 [53, 42]. The significant improvements manifest the effectiveness of MTR framework.

### 4.3  Ablation Study

We study the effectiveness of each component in MTR. For efficiently conducting ablation experiments, we uniformly sampled $20\%$ frames (about $97k$ scenes) from the WOMD training set according to their default order, and we empirically find that it has similar distribution with the full training set. All models are evaluated with marginal motion prediction metric on the validation set of WOMD.

**Effects of the motion decoder network.** We study the effectiveness of each component in our decoder network, including global intention localization, iterative refinement and local movement refinement. Table 3 shows that all components contributes remarkably to the final performance in terms of the official ranking metric mAP. Especially, our proposed static intention queries with intention points achieves much better mAP (*i.e.*, $+4.26\%$) than the latent learnable embeddings thanks to its mode-specific querying strategy, and both the iterative refinement and local movement refinement strategy continually improve the mAP from $30.59\%$ to $32.34\%$ by aggregating more fine-grained trajectory features for motion refinement.

**Effects of dense future prediction.** Table 3 shows that our proposed dense future prediction module significantly improves the quality of predicted trajectories (*e.g.*, $+1.78\%$ mAP), which verifies that future interactions of the agents' trajectories are important for motion prediction and our proposed strategy can learn such interactions to predict more reliable trajectories.

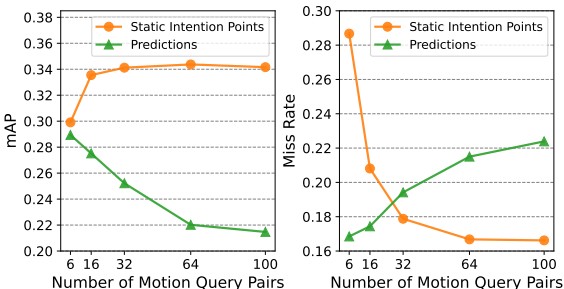

Figure 4: MTR framework with different number of motion query pairs, and two different colored lines demonstrate different strategies for selecting the positive mixture component during training process.

Table 4: Effects of local self-attention in transformer encoder. "#polyline" is the number of input map polylines used for context encoding, and a large number of polylines indicate that there is a larger map context around the interested agent. "OOM" indicates running out of memory.

| Attention | #Polyline | minADE ↓ | minFDE ↓ | MR ↓ | mAP ↑ |
|---|---|---|---|---|---|
| Global | 256 | 0.683 | 1.4031 | 0.1717 | 0.3295 |
| Global | 512 | 0.6783 | 1.4018 | 0.1716 | 0.3280 |
| Global | 768 | OOM | OOM | OOM | OOM |
| Local | 256 | 0.6724 | 1.3835 | 0.1683 | 0.3372 |
| Local | 512 | 0.6707 | 1.3749 | 0.1670 | 0.3392 |
| Local | 768 | **0.6697** | **1.3712** | 0.1668 | 0.3437 |
| Local | 1024 | 0.6757 | 1.3782 | **0.1663** | **0.3452** |

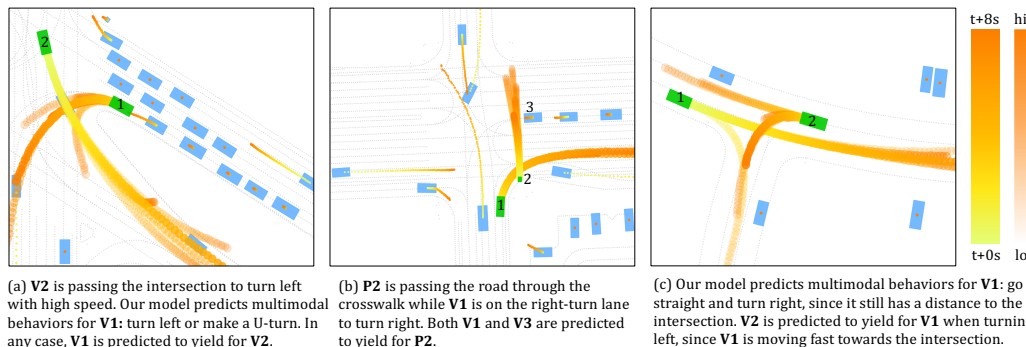

(a) **V2** is passing the intersection to turn left with high speed. Our model predicts multimodal behaviors for **V1**: turn left or make a U-turn. In any case, **V1** is predicted to yield for **V2**.

(b) **P2** is passing the road through the crosswalk while **V1** is on the right-turn lane to turn right. Both **V1** and **V3** are predicted to yield for **P2**.

(c) Our model predicts multimodal behaviors for **V1**: go straight and turn right, since it still has a distance to the intersection. **V2** is predicted to yield for **V1** when turning left, since **V1** is moving fast towards the intersection.

Figure 5: Qualitative results of MTR framework on WOMD. There are two interested agents in each scene (green rectangle), where our model predicts 6 multimodal future trajectories for each of them. For other agents (blue rectangle), a single trajectory is predicted by dense future prediction module. We use gradient color to visualize the trajectory waypoints at different future time step, and trajectory confidence is visualized by setting different transparent. Abbreviation: Vehicle (**V**), Pedestrian (**P**).

**Effects of local attention for context encoding.** Table 4 shows that by taking the same number of map polylines as input, local self-attention in transformer encoder achieves better performance than global attention (*i.e.*, +0.77% mAP for 256 polylines and +1.12% mAP for 512 polylines), which verifies that the input local structure is important for motion prediction and introducing such prior knowledge with local attention can benefit the performance. More importantly, local attention is more memory-efficient and the performance keeps growing when improving the number of map polylines from 256 to 1,024, while global attention will run out of memory due to its quadratic complexity.

**Effects of the number of motion query pairs with different training strategies.** As mentioned before, during training process, MTR and MTR-e2e adopt two different strategies for assigning positive mixture component, where MTR depends on static intention points (denoted as $\alpha$) while MTR-e2e utilizes predicted trajectories (denoted as $\beta$). Figure 4 investigates the effects of the number of motion query pairs under these two strategies, where we have the following observations: (1) When increasing the number of motion query pairs, strategy $\alpha$ achieves much better mAP and miss rate than strategy $\beta$. Because intention query points can ensure more stable training process since each intention query points is responsible to a specific motion mode. In contrast, strategy $\beta$ depends on unstable predictions and the positive component may randomly switch among all components, so a large number of motion query pairs are hard to be optimized with strategy $\beta$. (2) The explicit meaning of each intention query point also illustrates the reason that strategy $\alpha$ consistently achieves much better mAP than strategy $\beta$, since it can predict trajectories with more confident scores to benefit mAP metric. (3) From another side, when decreasing the number of motion query pairs, the miss rate of strategy $\alpha$ greatly increases, since a limit number of intention query points can not well cover all potential motions of agents. Conversely, strategy $\beta$ works well for a small number of motion query pairs since its queries are not in charge of specific region and can globally adapt to any region.

# 5 Conclusion

In this paper, we present MTR, a novel framework for multimodal motion prediction. The motion query pair is defined to model motion prediction as the joint optimization of global intention localization and local movement refinement. The global intention localization adopts a small set of learnable static intention queries to efficiently capture agent's motion intentions, while the local movement refinement conducts iterative motion refinement by continually probing fine-grained trajectory features. The experiments on both marginal and joint motion prediction challenges of large-scale WOMD dataset show that our approach achieves state-of-the-art performance.

**Limitations.** The proposed framework adopts an agent-centric strategy to predict multimodal future trajectories for one interested agent, leading redundant context encoding for multiple interested agents in the same scene. Hence, how to develop a framework that can simultaneously predict multimodal motion for multiple agents is one important future work. Besides, the rule-based post-processing can result in suboptimal predictions for minADE and minFDE metrics, and how to design a better strategy to produce a required number of future trajectories (e.g., 6 trajectories) from full multimodal predictions (e.g., 64 predictions) is also worth exploring for a more robust framework.

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
