# Motion Transformer with Global Intention Localization and Local Movement Refinement

**Shaoshuai Shi, Li Jiang, Dengxin Dai, Bernt Schiele**
Max Planck Institute for Informatics, Saarland Informatics Campus
{sshi, lijiang, ddai, schiele}@mpi-inf.mpg.de

# Appendix

## 1 Implementation Details

**More architecture details.** We train a single model for predicting the future motion of all three categories (*i.e.*, Vehicle, Pedestrian, Cyclist), and each category has their own motion query pairs.

The input history state $A_{\text{in}}$ contains three types of information, including agent history motion state (*i.e.*, position, object size, heading angle and velocity), one-hot category mask of each agent and one-hot time embedding of each history time step. The polyline encoder for $A_{\text{in}}$ contains a three-layer MLP with feature dimension 256. The input map feature $M_{\text{in}}$ contains three types of information, including the position of each polyline point, the polyline direction at each point, and the type of each polyline. The polyline encoder for $M_{\text{in}}$ contains a five-layer MLP with feature dimension 64, where we adopt a smaller feature dimension since the number of map polylines is much larger than the number of agents. Both two polyline encoders are finally projected to 256 feature dimension with another linear layer separately.

For the dense future prediction module, we adopt a three-layer MLP with intermediate feature dimension 512 for predicting future position and velocity of all agents. For the prediction head in each decoder layer, we adopt a three-layer MLP with intermediate feature dimension 512, and the model weights are not shared across different decoder layers.

**The details of Gaussian regression loss.** Given the predicted Gaussian Mixture Models for a specific future time step, we adopt negative log-likelihood loss to maximum the likelihood of the agent's ground-truth position $(\hat{Y}_x, \hat{Y}_y)$ at this time step, and the detailed loss can be formulated as:

$$L_G = -\log \mathcal{N}_h(\hat{Y}_x - \mu_x, \sigma_x; \hat{Y}_y - \mu_y, \sigma_y; \rho) - \log(p_h) \tag{1}$$

$$= \log \sigma_x + \log \sigma_y + 0.5 \log(1 - \rho^2) + \frac{1}{2(1-\rho^2)} \left( \left(\frac{d_x}{\sigma_x}\right)^2 + \left(\frac{d_y}{\sigma_y}\right)^2 - 2\rho \frac{d_x d_y}{\sigma_x \sigma_y} \right) - \log(p_h),$$

where $d_x = \hat{Y}_x - \mu_x$, $d_y = \hat{Y}_y - \mu_y$, and $\mathcal{N}_h(\mu_x, \sigma_x; \mu_y, \sigma_y; \rho)$ is the selected positive Gaussian component for optimization. $p_h$ is the predicted probability of this selected positive Gaussian component, and we adopt cross entropy loss in the above equation to maximum the probability of selected positive Gaussian component.

**Final training losses.** Given the above definition of Gaussian regression loss, the final training loss of MTR framework is the sum of all the Gaussian regression loss in each decoder layer and the auxiliary regression loss with equal loss weights.

**The distribution of static intention points.** As mentioned in the paper, we adopt k-means clustering algorithm to generate 64 static intention points for each category. As shown in Figure 1, the generated intention points can well cover most motion intentions by jointly considering both the velocity and direction of future motion. For example, vehicle category has a series of intention points in the heading direction that actually model the same intent direction with different velocities.

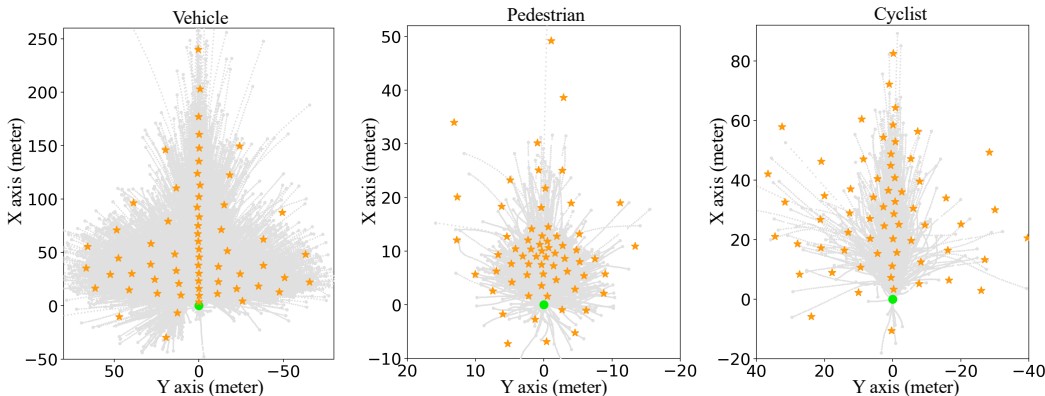

Figure 1: The distribution of static intention points for each category. The green point is the current position of the agent, and the intention points are shown as orange stars. The gray dotted lines indicate the distribution of ground-truth trajectory for each category, and note that only 10% ground-truth trajectories are drawn in the figure for better visualization.

## 2 Per-class Results of MTR Framework

As shown in Table 1 and 2, we report the per-category performance of our approach for both the marginal and joint motion prediction challenges of Waymo Open Motion Dataset [2] for reference.

Table 1: Per-class performance of marginal motion prediction on the validation and test set of Waymo Open Motion Dataset. †: The results are shown in *italic* for reference since the performance is achieved with model ensemble technique.

|  | Setting | Category | minADE ↓ | minFDE ↓ | Miss Rate ↓ | mAP ↑ |
|---|---|---|---|---|---|---|
| Test | MTR | Vehicle | 0.7642 | 1.5257 | 0.1514 | 0.4494 |
|  |  | Pedestrian | 0.3486 | 0.7270 | 0.0753 | 0.4331 |
|  |  | Cyclist | 0.7022 | 1.4093 | 0.1786 | 0.3561 |
|  |  | **Avg** | 0.6050 | 1.2207 | 0.1351 | 0.4129 |
| Val | MTR | Vehicle | 0.7665 | 1.5359 | 0.1527 | 0.4534 |
|  |  | Pedestrian | 0.3456 | 0.7260 | 0.0741 | 0.4364 |
|  |  | Cyclist | 0.7016 | 1.4134 | 0.1830 | 0.3594 |
|  |  | **Avg** | 0.6046 | 1.2251 | 0.1366 | 0.4164 |
|  | MTR-e2e | Vehicle | 0.6087 | 1.2161 | 0.1160 | 0.3708 |
|  |  | Pedestrian | 0.3046 | 0.6254 | 0.0696 | 0.3028 |
|  |  | Cyclist | 0.6346 | 1.2796 | 0.1845 | 0.2998 |
|  |  | **Avg** | 0.5160 | 1.0404 | 0.1234 | 0.3245 |

Table 2: Per-class performance of joint motion prediction on the validation and test set of Waymo Open Motion Dataset.

|  | Setting | Category | minADE ↓ | minFDE ↓ | Miss Rate ↓ | mAP ↑ |
|---|---|---|---|---|---|---|
| Test | MTR | Vehicle | 0.9793 | 2.2157 | 0.3833 | 0.2977 |
|  |  | Pedestrian | 0.7098 | 1.5835 | 0.3973 | 0.2033 |
|  |  | Cyclist | 1.0652 | 2.3908 | 0.5428 | 0.1102 |
|  |  | **Avg** | 0.9181 | 2.0633 | 0.4411 | 0.2037 |
| Val | MTR | Vehicle | 0.9675 | 2.1728 | 0.3722 | 0.3018 |
|  |  | Pedestrian | 0.7097 | 1.5522 | 0.3984 | 0.1892 |
|  |  | Cyclist | 1.0623 | 2.4357 | 0.5410 | 0.1065 |
|  |  | **Avg** | 0.9132 | 2.0536 | 0.4372 | 0.1992 |

## 3 Evaluation of Dense Future Prediction

As introduced in the paper, we propose the dense future prediction module to enable the future interactions among the agents, where the accuracy of the predicted trajectories for all neighboring agents is important for predicting better multimodal trajectories of our interested agent. As shown in

Table 3: The performance of dense future prediction on the validation set of Waymo Open Motion Dataset. The dense future prediction module predicts a single future trajectory (8 seconds into the future) for each agent in the validation set. ADE indicates average displacement error, and FDE indicates final displacement error. The FDE is evaluated based on the last time step of future 8 seconds, and the miss rate is evaluated based on FDE with different distance thresholds.

| Category | ADE ↓ | FDE ↓ | Miss Rate (thresh=2m) ↓ | Miss Rate (thresh=6m) ↓ |
|---|---|---|---|---|
| Vehicle | 0.8755 | 3.4929 | 0.3343 | 0.1984 |
| Pedestrian | 0.4998 | 1.9685 | 0.3317 | 0.0601 |
| Cyclist | 1.6565 | 6.3868 | 0.7697 | 0.3747 |
| **Avg** | 1.0106 | 3.9494 | 0.4786 | 0.2111 |

Table 4: The performance comparison with the top-10 submissions on the test set leaderboard of Argoverse2 dataset. $K$ is the number of predicted trajectories for calculating the evaluation metrics.

| Method | Miss Rate ($K$=6) ↓ | Miss Rate ($K$=1) ↓ | brier-minFDE ($K$=6) ↓ |
|---|---|---|---|
| MTR (Ours) | **0.15** | **0.58** | 1.98 |
| TENET [5] | 0.19 | 0.61 | **1.90** |
| OPPred | 0.19 | 0.60 | 1.92 |
| Qml | 0.19 | 0.62 | 1.95 |
| GANet | 0.17 | 0.60 | 1.97 |
| VI LaneIter | 0.19 | 0.61 | 2.00 |
| QCNet | 0.21 | 0.60 | 2.14 |
| THOMAS [3] | 0.20 | 0.64 | 2.16 |
| HDGT [4] | 0.21 | 0.66 | 2.24 |
| GNA | 0.29 | 0.71 | 2.45 |
| vilab | 0.29 | 0.71 | 2.47 |

Table 3, our simple dense future prediction module can predict accurate future trajectory for all agents, where the average miss rate with $6.0m$ distance threshold is $21.11\%$ for future motion prediction of 8 seconds. It's worth noting that this miss rate is achieved with a single predicted future trajectory for each agent. Although the trajectory generated by the dense future prediction module is still not as accurate as that of the decoder network, they can already benefit the multimodal motion prediction of our interested agent by enabling the future interactions of the agents.

## 4 Performance Comparison on Argoverse 2 Dataset

The Argoverse 2 Motion Forecasting Dataset [6] is another large-scale dataset for motion prediction, which contains 250,000 scenarios for training and validation. The model needs to take the history five seconds of each scenarios as niput, and predict the six-second future trajectories of one interested agent, where HDMap is always available to provide map context information. To train our model on this dataset, we adopt the same hyper-parameters as in Waymo dataset, except that the model takes five-second history trajectories as input and needs to predict six-second future trajectories.

We compare out approach with the top-10 submissions on the leaderboard of Argoverse2 dataset [1], where most submissions are tailored for Argoverse2 Motion Forcasting Competition 2022 and are highly competitive. As shown in Table 4, our MTR framework achieves new state-of-the-art performance with remarkable gains on the miss-rate related metrics, demonstrating the great generalizability and robustness of our approach.

Table 5: Effects of the number of neighbors for local self-attention in transformer encoder.

| #neighbors | minADE ↓ | minFDE ↓ | MR ↓ | **mAP** ↑ |
|---|---|---|---|---|
| 4 | **0.6677** | 1.3724 | 0.1672 | 0.3405 |
| 8 | 0.6681 | **1.3673** | 0.1670 | 0.3428 |
| 16 | 0.6697 | 1.3712 | **0.1668** | **0.3437** |
| 32 | 0.6701 | 1.3763 | 0.1678 | 0.3416 |
| 64 | 0.6727 | 1.3756 | 0.1687 | 0.3367 |

Table 6: Effects of the number of map polylines for dynamic map collection.

| #Polyline | minADE ↓ | minFDE ↓ | MR ↓ | **mAP** ↑ |
|---|---|---|---|---|
| 32 | 0.6735 | 1.3847 | 0.1701 | 0.3317 |
| 64 | 0.6699 | **1.3650** | 0.1672 | 0.3386 |
| 128 | **0.6697** | 1.3712 | 0.1668 | **0.3437** |
| 256 | 0.6704 | 1.3729 | **0.1665** | 0.3396 |

# 5 More Ablation Studies

**Effects of the number of neighbors for local self-attention.** As shown in Table 5, using 4 neighbors in local self-attention already achieves good performance in terms of mAP, and the mAP metric keeps growing when increasing the number of neighbors from 4 to 16. However, when we increase the neighbors to 64 for conducting local self-attention, the performance drops a bit (*i.e.*, from $34.37\%$ to $33.67\%$). It demonstrates that a small number of neighbors can better maintain the local structure of input elements and is easier to be optimized for achieving better performance, and a small number of neighbors is also much more computational- and memory-efficient than a larger number of neighbors for conducting self-attention.

**Effects of the number of polylines for dynamic map collection.** Table 6 shows that increasing the number of collected map polylines from 32 to 128 can constantly improve the mAP metric by retrieving trajectory-specific map features with larger receptive field. However, the performance drops a bit (-$0.41\%$ mAP) when collecting 256 map polylines for refining the trajectory, which shows that a larger number of collected map polylines may involve more noise and can not provide accurate trajectory-specific map features for refinement.

Table 7: Effects of the number of decoder layers.

| #decoders | minADE ↓ | minFDE ↓ | MR ↓ | mAP ↑ |
|---|---|---|---|---|
| 3 | 0.6717 | 1.3796 | 0.1686 | 0.3360 |
| 6 | 0.6697 | 1.3712 | 0.1668 | **0.3437** |
| 9 | **0.6658** | **1.3621** | **0.1661** | **0.3437** |

Table 8: Effects of the distribution of static intention points.

| #Distribution | minADE ↓ | minFDE ↓ | MR ↓ | mAP ↑ |
|---|---|---|---|---|
| uniform grids | 0.7214 | 1.5563 | 0.1970 | 0.3178 |
| k-means | **0.6697** | **1.3712** | **0.1668** | **0.3437** |

**Effects of the number of decoder layers.** Table 7 shows that increasing the number of decoder layers can constantly improve the performance, which demonstrates that our stacked transformer decoder layers can iteratively refine the predicted trajectories by continually aggregating more accurate trajectory-specific features. By default, we adopt 6 decoder layers by considering the trade-off between the performance and the efficiency of MTR framework.

**Effects of the distribution of static intention points.** As introduced in the paper, we adopt k-means algorithm to generate 64 static intention points as the basis of our motion query pairs. We ablate another simple uniform sampling strategy to generate the same number of static intention points for each category, where we uniformly sample $8 \times 8 = 64$ static intention points by considering the range of trajectory distribution of each category (see Figure 1). As shown in Table 8, the performance drops significantly when replacing k-means algorithm with the uniform sampling for generating static intention points. It verifies that our k-means based algorithm can produce better distribution of static intention points, which can capture more accurate and more complete future motion intentions with a small number of static intention points. Figure 1 also demonstrates the effectiveness of our default static intention points for capturing multimodal motion intentions.

# 6 Qualitative Results

We provide more qualitative results of our MTR framework in Figure 2.

# 7 Notations

As shown in Table 9, we provide a lookup table for notations in the paper.

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

Table 9: Lookup table for notations in the paper.

| | |
|---|---|
| $A_{\text{in}}$ | input history motion state of agent |
| $M_{\text{in}}$ | input map features with polyline representation |
| $C_a$ | input feature dimension of agent's state |
| $N_a$ | number of agents |
| $t$ | number of given history frames |
| $N_m$ | number of map polylines |
| $n$ | number of points in each map polyline |
| $A_{\text{past}}$ | agent features after polyline encoder |
| $A_{\text{future}}$ | encoded future features of the densely predicted trajectories for all agents |
| $A$ | agent features in transformer encoder |
| $M$ | map features in transformer encoder |
| $D$ | hidden feature dimension of transformer |
| PE | function of sinusoidal position encoding |
| $\kappa$ | function of k-nearest neighbor algorithm |
| $T$ | number of future frames to be predicted |
| $S_i$ | predicted future position and velocity of all agents at time step $i$ |
| $I$ | static intention points |
| $Q_I$ | static intention query |
| $Q_S^j$ | dynamic searching query in $j$-th decoder layer |
| $Y_{1:T}^j$ | predicted future trajectories in $j$-th decoder layer |
| $L$ | number of map polylines collected along the predicted trajectory |
| $C^{j-1}$ | input query content features of $j$-th decoder layer |
| $C_{\text{sa}}^j$ | query content features after self-attention of $j$-th decoder layer |
| $C_A^j$ | query agent features after cross-attention of $j$-th decoder layer |
| $C_M^j$ | query map features after cross-attention of $j$-th decoder layer |
| $\alpha$ | function of dynamic map collection |
| $Z_{1:T}^j$ | predicted GMM parameters in $j$-th decoder layer |
| $p_k$ | predicted probability of $k$-th component in GMM |
| $\mathcal{N}_k$ | function of $k$-th component in GMM |
| $\mathcal{K}$ | number of components for each GMM |
| $P_i^j(o)$ | predicted occurrence probability of the agent at spatial position $o$ and time step $i$ |