# OpenReview forum: "Motion Transformer with Global Intention Localization and Local Movement Refinement"
_NeurIPS.cc/2022/Conference — NeurIPS 2022 Accept_

### Official Review · Reviewer_Xhm6 · 2022-07-11

**Rating:** 7
**Confidence:** 5
**Soundness:** 3 good
**Presentation:** 3 good
**Contribution:** 3 good

**Summary:**

In this paper, a Motion TRansformer (MTR) framework is proposed for the motion prediction task, including marginal and joint motion prediction. Specifically, motion query pairs are designed for global intention localization and local movement refinement, which takes advantage of both goal-based methods and the regression methods. Experiments on Waymo Open Dataset indicate the effectiveness of the proposed method.

**Questions:**

Concerns:
1. Overall, it is a paper for motion prediction challenge with promising results and clear presentations. I assume it is the initial version of the final 1st place work [1], sharing very similar methods and almost the same framework, thus I would encourage authors to update the paper with [1].
2. The most concern is that all the strategies are well-studied in this community, i.e., the transformer idea, scene context and interaction modeling, etc. It would be better if more insights could be included for model design, and highlight the differences/contributions.
3. In Line 134-135, how to pad zeros for trajectories that have less than $t$ frames? In the front/back? Some details are missing here.
4. Some missing references when referring to future interactions, some works [2-4] have been proposed to argue the future interaction point and to consider such future interactions.

Minors:
1. Line 56, I believe it is 'empirical' instead of 'experimentally'.
2. In References, the names of conferences or journals, i.e. abbreviation or full name, are not consistent.

I am happy to improve my rating if these concerns could be addressed.


[1] https://storage.googleapis.com/waymo-uploads/files/research/MotionPred/MotionPrediction_MTRA.pdf
[2] EvolveGraph: Multi-Agent Trajectory Prediction with Dynamic Relational Reasoning. NeurIPS 2020
[3] Tra2Tra: Trajectory-to-Trajectory Prediction With a Global Social Spatial-Temporal Attentive Neural Network. ICRA 2021
[4] StarNet: Pedestrian Trajectory Prediction using Deep Neural Network in Star Topology. IROS 2019



**Limitations:**

There is a limitation section in the main body of the paper.

**Strengths And Weaknesses:**

Strengths:
- Strong motivation and well-organized.
- Promising results.


Weaknesses: (See Questions for details)
- Some insights/details are not clear.
- Lack of some experiments.

---

> ### Author Response · Authors · 2022-08-02
> **Response to Reviewer Xhm6**
>
> We sincerely thank the reviewer for providing thoughtful review and constructive suggestions. We answer the questions as follows.
>
> ****
>
> **Q1**: "I assume it is the initial version of the final 1st place work [1], sharing very similar methods and almost the same framework, thus I would encourage authors to update the paper with [1]."
>
> **A1**:
> Thanks for the valuable suggestion. We will follow the reviewer's suggestion and update the paper in the revised version.
>
> ****
>
> **Q2**: "The most concern is that all the strategies are well-studied in this community" and "It would be better if more insights could be included for model design, and highlight the differences/contributions."
>
> **A2**:
> Thanks for the constructive suggestion. We discuss more insights about the model design and highlight our differences and contributions in the following, which will be included in the revised version:
>
> * Our key contribution lies in the design of the motion decoder network. Indeed, we agree with the reviewer that there are already some works focusing on exploring the transformer idea, scene context and interaction modeling (*e.g.*, SceneTransformer [30], MultiPath++ [40], etc.). However, most of existing works focus on the model design of scene context encoding and agent interaction modeling, and how to design a better motion decoder for multimodal motion prediction is still underexplored.
> * Hence, we propose a transformer-based motion decoder network with a novel concept of motion query pair, where each motion query pair is bound with a fixed intention point and takes charge of generating the future trajectory of a specific motion mode.
> The insight is that modeling different motion modes with different motion query pairs facilitates better multimodal motion prediction and makes the predictions more explainable, since each motion query pair has explicit correspondence with a specific motion mode and predicts their own confidence/trajectory respectively.
> Moreover, our approach can predict very different motion modes of multiple categories (*e.g.*, vehicle, pedestrian and cyclist, etc.) in a single model, because the motion query pair utilizes shared MLP to encode the intention points, so that we just need to set different sets of prior intention points for different categories to get their corresponding motion query pairs.
> * We propose to iteratively refine the predicted trajectory in motion decoder network by continually probing trajectory-specific features in different decoder layers, which is novel and hasn't been explored in motion prediction field (also endorsed by Reviewer 3fr6). The insight is that the predicted future trajectory should be consistent with its local context environment, especially the HD map context. Hence we introduce dynamic map collection module to force the model to constantly focus on the latest context environment around the predicted trajectories, so as to refine them with fine-grained features.
> * Compared with previous goal-based methods [51,18],  our approach eliminates the dependence of dense goal candidates (*e.g.*, about $10k$ in Waymo dataset) by introducing a small set of learnable motion query pairs (*e.g.*, 64 in Waymo dataset), which makes our approach more efficient than these methods.
> Compared with other state-of-the-art methods (*e.g.*, SceneTransformer [30], MultiPath++ [40]), our approach introduces intention priors through motion query pairs, which can predict better multimodal motion by separately considering each explicit motion mode. In contrast, SceneTransformer [30] predicts various motion modes from the same agent feature and MultiPath++ uses implicit and latent anchor embeddings.
>
> ****
>
> **Q3**: "In Line 134-135, how to pad zeros for trajectories that have less than frames?"
>
> **A3**: We apologize for the confusion. The history motion state $A_{in}$ is represented with full $t$ frames according to their frame positions, where the zeros are padded at the position of the missing frames, since these invalid frames may be intermediate frames of $t$ frames due to temporary occlusion. We will clarify this in the revised version.
>
> ****
>
>
> **Q4**: "Some missing references when referring to future interactions, some works [2-4] have been proposed to argue the future interaction point and to consider such future interactions."
>
> **A4**:
> Thanks for pointing out these valuable references. We will cite and discuss these works in the revised version.
>
> ****
>
> **Q5**: "Line 56, I believe it is 'empirical' instead of 'experimentally'."
>
> **A5**:
> Thanks for pointing out this. We will revise it according to the reviewer's suggestion.
>
> ****
>
> **Q6**: "In References, the names of conferences or journals, i.e. abbreviation or full name, are not consistent."
>
> **A6**: Thanks for pointing out this issue. We will revise the reference to make it consistent.

---

> > ### Comment · Reviewer_Xhm6 · 2022-08-04
> > **Response to Author**
> >
> > Thank you for your answers. Most of my concerns are well-addressed.
> >
> > I am planning to increase the score. Once the revised paper is uploaded, I will increase.

---

> > > ### Author Response · Authors · 2022-08-06
> > > **Thanks for your positive feedback!**
> > >
> > > Thank you very much for the positive feedback!
> > >
> > > According to your valuable suggestions, we have revised and uploaded the paper and appendix.
> > > We are happy to further update the paper / appendix if there are any other suggestions.
> > > Thanks!

---

> > > > ### Comment · Reviewer_Xhm6 · 2022-08-08
> > > > **Thank you for your revised paper**
> > > >
> > > > Thank you for uploading the revised paper. It's a great work and I increased my score to 7.

---

### Official Review · Reviewer_4kMs · 2022-07-11

**Rating:** 7
**Confidence:** 4
**Soundness:** 3 good
**Presentation:** 2 fair
**Contribution:** 3 good

**Summary:**

The work addresses the motion forecasting problem for autonomous driving. The authors introduce a transformer-based framework (MTR) that works in the following ways, as highlighted in the contribution section of the paper:

- Separates the modeling of the global intention from local movement refinement in the trajectories in the transformer framework. The predictor is inspired by DETR.
- Interaction modeling between agents via an auxiliary dense prediction task, essentially letting the model to predict the future directly (I didn't fully understand that part... see below)
- SOTA results on the Waymo dataset. Ranked #1 among results without using ensembles.

Since I don't understand the 2nd contribution, I cannot recommend accept at this point. Looking forward to understanding it post-rebuttal!

POST REBUTTAL UPDATE: I understand the contribution better now. In line with other reviews' feedback, I'll change the rating from 4 -> 7, and recommend an accept.


**Questions:**

line 73: Why not also using ensemble to boost the accuracy a bit to achieve top results?

Figure 1: The figure is far from self explaining. What do the small road illustrations above the dense future prediction and dynamic map construction say? What are K, T, N and D (I know its in the text, but it doesn't hurt to say it here)? Where do the input "predicted trajectories", "map query" and "query content feature" come from? The figure is way too complicated, and I don't feel rewarded for looking closely....

Eq 2, ln 164: There are equations like   X = f(X), e.g. A = MLP([A, A_t])   what does this even mean? please use subscripts to make it clear that the variable in LHS and RHS are not the same.

ln 237: where does the auxiliary loss L_a come from? just write equations and not words. And if its just an L1 loss against the GT. I don't understand the intuition why it helps with interaction. It seems like its just like a multi-head network where you supply the GT at several layers of the network.

**Limitations:**

Yes, but it didn't hit the mark.

The limitation section is about high-level limitation, like "the method is not great for long tail behaviors", not a wish list of future works.

**Strengths And Weaknesses:**

Strengths:

Strong results: Significant improvement over the SOTA. I checked the leaderboard on Waymo Open Dataset, and verified the claims.

Somewhat novelty: Using DETR for motion forecasting is not that new anymore. I reviewed several papers for CVPR'22 and ECCV'22 that contained similar ideas. But those should be considered concurrent work. Additionally, the hierarchical separation of intent vs. fine control adds an additional hint of novelty, though the hierarchical separation by itself is not novel either [9, 51].

Good ablation: The paper contains ablation for all the contributions and novelties in their methods. This is great!

Weaknesses:

- Unclear writing: Abused notations, missing descriptions of variables, figures that are not self-contained. See below under "questions".

- Insufficient experiments (minor): The only results are on the Waymo dataset. There are other popular datasets such as Argoverse and nuScenes. Having strong results on a secondary dataset would make the representation a lot stronger.

---

> ### Author Response · Authors · 2022-08-02
> **Response to Reviewer 4kMs (1/2)**
>
> We sincerely thank the reviewer for providing thoughtful review and constructive suggestions. We answer the questions as follows.
>
> ****
>
> **Q1**: "Somewhat novelty: Using DETR for motion forecasting is not that new anymore. I reviewed several papers for CVPR'22 and ECCV'22 that contained similar ideas. But those should be considered concurrent work. Additionally, the hierarchical separation of intent vs. fine control adds an additional hint of novelty, though the hierarchical separation by itself is not novel either [9, 51]."
>
> **A1**:
> We appreciate the reviewer's acknowledgement and recognition of some novelties of the proposed framework.
> Here we would like to highlight our core contributions and differences compared to existing works:
>
> * The key contribution of our approach lies in the design of motion decoder network.
> To model multimodal motion modes of the agent, we propose a novel concept of motion query pair in the decoder network, where each motion query pair is bound with a fixed intention point and takes charge of generating the future trajectory of a specific motion mode.
> This independent modeling of different motion modes facilitates better multimodal motion prediction. The major differences compared to existing works are discussed below:
>     * Compared with other DETR related motion predictors like MultiPath++ [40]: they generally consider queries as latent anchor embeddings while our motion query pair has explicit meaning and is more explainable. For example, in our framework, each motion query pair corresponds to a specific local region around its prior intention point, and the position of this intention point can jointly consider both the motion direction (*e.g.*, turn left, turn right and go straight, etc.) and moving distance (*e.g.*, velocity) by considering this local region as agent's destination.
>     * Compared with TNT/DenseTNT [51,18]: they generally depend on a very large number of goal candidates (*e.g.*, about 10k for Waymo dataset) to achieve good performance, while our approach is much more efficient by only adopting a small set of motion query pair (*e.g.*, 64 for Waymo dataset).
>     * Compared with MultiPath [9]: Unlike MultiPath which uses anchor trajectories, our intention point modeling can better handle various possible trajectories for the same intention, since our learnable motion query pairs can dynamically predict different feasible trajectories based on the context information.
> * We propose to iteratively refine the predicted trajectory by continually probing trajectory-specific features with stacked transformer decoder layers, which is novel and hasn't been explored in motion prediction field (also endorsed by Reviewer 3fr6).
>
>
> We are happy to compare and discuss more related works in the revised version, and we are also happy to answer any further questions.
>
> ****
>
> **Q2**: "Unclear writing: Abused notations, missing descriptions of variables, figures that are not self-contained."
>
> **A2**: We apologize for the confusion. We will follow the reviewer's suggestions to utilize different notations for LHS and RHS of Eq.(2) / line-164, revise the figure and add more descriptions to the caption to make it self-explaining.
>
> ****
>
> **Q3**: "Insufficient experiments (minor)" and "Having strong results on a secondary dataset would make the representation a lot stronger."
>
> **A3**:
> Thanks for the valuable suggestion. As suggested by the reviewer,
> we further evaluate our approach on the latest large-scale Argoverse2 dataset.
> Given the limited time and resources during the rebuttal period, we directly train our approach with the same hyper-parameters as in Waymo dataset.
> As shown in the following table, we compare our approach with the top-10 submissions on the leaderboard of Argoverse2 dataset [A] (note that these submissions are highly competitive since most of them are tailored submissions for this year's Argoverse2 competition in June).
> The experiments show that our preliminary results already achieve new state-of-the-art performance with remarkable gains on two miss rate related metrics, demonstrating the great generalizability and robustness of our approach.
> We will follow the reviewer's suggestion and add the evaluation of this secondary dataset in the revised version.
>
> | Method | Miss Rate (K=6) | Miss Rate (K=1) | brier-minFDE (K=6) |
> |:---|:---:|:-----:|:--:|
> | MTR (Ours)| **0.15** | **0.58** | 1.98|
> | TENET| 0.19 | 0.61  | **1.90**|
> | OPPred| 0.19 |0.60 | 1.92 |
> | Qml| 0.19 | 0.62 |1.95|
> | GANet| 0.17 |0.60 |1.97 |
> | VI LaneIter| 0.19 | 0.61 | 2.00 |
> | QCNet |0.21 | 0.60| 2.14 |
> | GOHOME Scalar  | 0.20 | 0.64 |2.16|
> | HDGT  | 0.21	| 0.66 | 2.24|
> | GNA  | 0.29 | 0.71 | 2.45|
> | vilab  | 0.29	| 0.71 | 2.47 |
> The performance comparison on the test set leaderboard of Argoverse2 dataset. Note that K is the number of predicted trajectories for calculating the evaluation metrics.
>
> \[A\]: https://eval.ai/web/challenges/challenge-page/1719/leaderboard/4098
>
> ****

---

> > ### Author Response · Authors · 2022-08-02
> > **Response to Reviewer 4kMs (2/2)**
> >
> > **Q4**: "line 73: Why not also using ensemble to boost the accuracy a bit to achieve top results?"
> >
> > **A4**: Because different ensemble strategies may lead to very different performance gains, causing unfair comparison of different models. Therefore, in the introduction section, we only report the ensemble-free performance to demonstrate the effectiveness of the model itself.
> > Actually, Table 1 shows that our ensemble-free performance already outperforms the state-of-the-art model ensemble performance of MultiPath++ [40] in terms of mAP.
> > Moreover, we have also provided a model ensemble performance (*i.e.* MTR-ens)  in Table 1 by utilizing a simple ensemble strategy, which achieves much better performance than our default setting (43.23\% vs. 41.64\% in terms of mAP).
> >
> > ****
> >
> > **Q5**: "Figure 1: The figure is far from self-explaining."
> >
> > **A5**: Thanks for pointing out this issue. We will revise the figure and caption to make it self-explaining. The caption will be updated as something like:
> >
> > ``The architecture of MTR framework for multimodal motion prediction. $K$ is the number of motion query pairs, $T$ is the number of future frames to be predicted, $D$ is the hidden feature dimension and $N$ is the number of transformer decoder layers. The predicted trajectories, motion query pairs, and query content features are the outputs from last decoder layer and will be taken as input to current decoder layer. For the first decoder layer, both two components of motion query pair are initialized as predefined intention points, the predicted trajectories are replaced with the intention points for initial map collection, and query content features are initialized as zeros.''
> >
> > ****
> >
> > **Q6**: Some confusion about "Eq 2, ln 164", and "please use subscripts to make it clear that the variable in LHS and RHS are not the same."
> >
> > **A6**: We apologize for the confusion and thank the reviewer for the valuable suggestion. We will follow the suggestion to use subscript to make LHS and RHS clear.
> >
> > ****
> >
> > **Q7**: "ln 237: where does the auxiliary loss $L_a$ come from?" and "if it's just an L1 loss against the GT. I don't understand the intuition why it helps with interaction."
> >
> > **A7**: We apologize for the confusion and provide the following clarifications:
> >
> > * **Auxiliary loss**: $L_a$ is calculated by L1 regression loss, which only optimizes the prediction of each agent to predict a single future trajectory as in Eq.(3) and does not enable the interaction of the agents. We will add the loss equation and clarify this in the revised version.
> > * **How dense prediction task helps interaction**:
> > The dense future prediction task can predict future trajectories of all agents within the same scene, which are encoded and then taken as input to the transformer decoder network for providing additional future context information.
> > This future context information helps the interaction between the interested agent (*i.e.*, the agent at the center of the normalized input coordinate) and other agents, and facilitates the model to predict more scene-compliant future trajectories for the interested agent.
> > For example, if the interested agent has interactions with one or multiple surrounding agents, the dense future prediction module can predict potential future trajectories of these surrounding agents as additional future context information, which is input to the transformer decoder network, so that the model can predict better future trajectories for the interested agent by considering the consistency between its predicted trajectories and these potential future trajectories of the surrounding agents.
> > The experiments in Table 3 demonstrate that the dense future prediction module can effectively improve the performance from 32.84\% to 34.37\% in terms of mAP.
> >
> > ****
> >
> > **Q8**: "The limitation section is about high-level limitation."
> >
> > **A8**: We will revise the limitation section and add more discussion as follows:
> >
> > "The proposed framework adopts an agent-centric strategy to predict multimodal future trajectories for one interested agent, which leads to redundant context encoding if there are multiple interested agents in the same scene. Although the dense future prediction module partially compensates for this limitation, it can only predict a single future trajectory for each agent. Hence, how to develop a joint motion prediction framework that can simultaneously predict multimodal motion for multiple agents is one important future work.
> > Besides, the rule-based NMS post-processing can result in suboptimal predictions for minADE and minFDE metrics, and how to develop a learning-based module to produce a required number of future trajectories (*e.g.*, 6 trajectory) from full multimodal predictions (*e.g.*, 64 predictions) is also worth exploring for a more robust framework."

---

> > > ### Comment · Reviewer_4kMs · 2022-08-06
> > > **Thank you**
> > >
> > > Thank you for the detailed feedback. I understand the 2nd contribution better now. Please do improve the readability of the paper itself.
> > >
> > > I'll recommend accept. Thank you!

---

> > > > ### Author Response · Authors · 2022-08-06
> > > > **Thank you for your positive feedback!**
> > > >
> > > > Thank you very much for the positive feedback!
> > > >
> > > > According to your valuable suggestions, we have revised and uploaded the paper and appendix.
> > > > We will continue to polish the paper for better readability.
> > > > Thanks!

---

### Official Review · Reviewer_3fr6 · 2022-07-11

**Rating:** 8
**Confidence:** 5
**Soundness:** 4 excellent
**Presentation:** 3 good
**Contribution:** 4 excellent

**Summary:**

This paper proposes an decoder method for motion prediction task, which refines different modes with the static prior and dynamic attention. Its performance on the Waymo Open Motion Dataset is impressive, which demonstrates the effectiveness of iteratively refine the prediction (similar to DETR/DAB-DETR).

**Questions:**

#### Question
* The PE module is first introduced in the NLP field so that tokens' position information could be considered (Transformer is a set operator which ignores the postion information).  However, in the field of motion prediction, the feature itself includes the postion information (coordinate). I am wondering why the proposed method uses the PE + MLP instead of just MLP. Is there any experiments about its effectiveness?
* In equation (6), it uses C^{j-1}. How to initialize it in the first layer is not described.
* I noticed that Waymo Motion Challenge 2022 recently finished. Authors may discuss about those top-ranked methods including golfer, MPA, HDGT, etc, which could mkae the paper more up-to-date.

**Limitations:**

The limitation part is ok.

**Strengths And Weaknesses:**

Strengths
* The idea of iteratively refining the prediction by Transformer is novel in the motion prediction area.
* The performance on Waymo Motion Dataset is great.
* The ablation experiments is well-organized and convincing

Weakness
* The proposed method is only evaluated on one dataset. It would be more convincing if the experiments could be done on other large scale datasets. However, considering the Waymo Open Motion is fairly large, I think it is okay if there is no enough time/resource to try other datasets.
* Some parts of the proposed methods is not clearly described in the manuscript, which is understandable considering the space limit.
* The proposed method has similarities with the DAB-DETR in the objection detection area. I think it would better position this work if the authors could discuss about the proposed method's relation with recent DETR related works.

---

> ### Author Response · Authors · 2022-08-02
> **Response to Reviewer 3fr6 (1/2)**
>
> We sincerely thank the reviewer for providing thoughtful review and constructive suggestions. We answer the questions as follows.
>
> ****
>
> **Q1**:
> "It would be more convincing if the experiments could be done on other large scale datasets." and "it is okay if there is no enough time/resource to try other datasets."
>
> **A1**:
> Thanks for the valuable suggestion.
> We evaluate our proposed approach on the latest Argoverse2 dataset, which is another large-scale dataset for motion prediction.
> Given the limited time and resources during the rebuttal period, we directly train our framework with the same hyper-parameters as in Waymo dataset.
> As shown in the following table, we compare our approach with the top-10 submissions on the leaderboard of Argoverse2 dataset [A] (note that these submissions are highly competitive since most of them are tailored submissions for this year's Argoverse2 competition in June).
> The experiments show that our preliminary results already achieve new state-of-the-art performance with remarkable gains on the miss-rate related metrics, demonstrating the great generalizability and robustness of our approach.
> We will include the evaluation of our approach on this new dataset in the revised version.
>
> | Method | Miss Rate (K=6) | Miss Rate (K=1) | brier-minFDE (K=6) |
> |:---|:---:|:-----:|:--:|
> | MTR (Ours)| **0.15** | **0.58** | 1.98  |
> | TENET   | 0.19 | 0.61  | **1.90** |
> | OPPred | 0.19 |	0.60 | 1.92 |
> | Qml  | 0.19 | 0.62 | 1.95 |
> | GANet  | 	0.17 |	0.60 | 1.97 |
> | VI LaneIter   | 	0.19 | 0.61 | 2.00 |
> | QCNet |   0.21 | 0.60	| 2.14 |
> | GOHOME Scalar  | 	0.20 | 0.64 | 2.16 |
> | HDGT  | 0.21	| 0.66 | 2.24 |
> | GNA  | 0.29 | 0.71 | 2.45 |
> | vilab  | 0.29	| 0.71 | 2.47 |
> The performance comparison on the test set leaderboard of Argoverse2 dataset. Note that K is the number of predicted trajectories for calculating the evaluation metrics.
>
> \[A\]: https://eval.ai/web/challenges/challenge-page/1719/leaderboard/4098
>
> ****
>
> **Q2**: "It would better position this work if the authors could discuss about the proposed method's relation with recent DETR related works."
>
> **A2**:
> Thanks for the constructive suggestion. We will add a discussion section about this in the revised version.
> Here we discuss the relation and differences between DETR related works and our approach to better demonstrate our specific model design for motion prediction task:
>
> * Indeed, as mentioned in L109, our work is inspired by DETR [4] and its follow-up works like DAB-DETR [23].
> Specifically, both DAB-DETR [23] and our work utilize sinusoidal positional encoding and MLP to explicitly encode a spatial anchor point as positional query in transformer decoder layer.
> However, we adopt different definition and usage of the positional query to make our model more explainable for multimodal motion prediction. Concretely, instead of using dynamic learnable query as in DETR related works, we propose to use motion query pair as positional query, where each motion query pair is bound with a fixed intention point, taking charge of predicting the future trajectory of a specific motion mode (*e.g.*, turn left, turn right and go straight, etc.) towards this intention point.
> This independent modeling of different motion modes enables our framework to predict better multimodal trajectories.
> In contrast, the positional query in DETR related works may be dynamically associated with different ground-truth objects and does not have explicit and stable correspondence.
> * Besides, during the iterative refinement with transformer decoder layers, we propose dynamic map collection to constrain the input map to a deformable region along the predicted trajectory for fine-grained trajectory refinement. This explicit attention constraint is novel and hasn't been explored in DETR related works.
>
> ****

---

> > ### Author Response · Authors · 2022-08-02
> > **Response to Reviewer 3fr6 (2/2)**
> >
> > **Q3**: "In the field of motion prediction, the feature itself includes the postion information (coordinate). I am wondering why the proposed method uses the PE + MLP instead of just MLP. Is there any experiments about its effectiveness?"
> >
> > **A3**:
> > Thanks for the valuable question.
> > The PE module has been utilized in three components of the proposed framework, which are ablated in the following to investigate their individual effectiveness:
> >
> > * Using PE + MLP to encode motion query pair in Eq.(4,5): Here the PE module aims to map raw point coordinate to a sinusoidal positional embedding space, so as to be consistent with the sinusoidal positional embedding of keys. The experiments ($1^{st}$ and $2^{nd}$ rows) show that removing PE and applying MLP directly on raw coordinate lead to a slight performance decrease.
> > * Using PE in cross attention as in Eq.(5,7): The experiments ($1^{st}$ and $3^{rd}$ rows) show that the performance drops a lot if we remove all the positional embeddings in cross attention module. It demonstrates that although all features naturally contain position information, the key/value and query are still not well aligned since they are deeply encoded by different sub-networks (*i.e.*, context encoder for key/value and transformer decoder for query). Hence, the positional embedding plays an important role in cross attention to associate the input key/value and query.
> > * Using PE in self attention of transformer encoder as in Eq.(2): The experiments ($1^{st}$ and $4^{th}$ rows) show that removing the positional embedding in transformer encoder just slightly decreases the performance.
> > It is in line with the reviewer's argument: the positional embedding is not important in self attention since the input feature is the same for key/value/query and it already includes position information. However, the slight performance decrease also implies that adding the positional embedding to self attention can constantly augment the positional information and benefit the performance.
> >
> >
> > | Setting | Miss Rate (low) | mAP (high) |
> > |:--|:--:|:--:|
> > | MTR | **0.1668** | **0.3437** |
> > | Remove PE in Motion Query Pair - Eq.(4,5) | 0.1730 | 0.3328|
> > | Remove PE in Cross Attention - Eq.(5,7) | 0.3304 | 0.2075 |
> > | Remove PE in Encoder - Eq.(2) | 0.1707 | 0.3424|
> >
> >
> > ****
> >
> > **Q4**: "In equation (6), it uses $C^{j-1}$. How to initialize it in the first layer is not described."
> >
> > **A4**: Thanks for pointing out this. We follow the common setting in DETR related works to initialize $C^0$ as a zero tensor. We will clarify it in the revised version.
> >
> > ****
> >
> > **Q5**: "I noticed that Waymo Motion Challenge 2022 recently finished. Authors may discuss about those top-ranked methods including golfer, MPA, HDGT, etc, which could make the paper more up-to-date."
> >
> > **A6**:
> > Thanks for the constructive suggestion. We will compare and discuss these latest references in the revised version.

---

### Official Review · Reviewer_b34B · 2022-07-12

**Rating:** 5
**Confidence:** 4
**Soundness:** 3 good
**Presentation:** 2 fair
**Contribution:** 3 good

**Summary:**

This paper proposes a transformer-based method for vehicle trajectory forecasting. It proposes to combine the tasks of global localization and local refinement for more accurate trajectory forecasting. On the structure side, the authors design a mechanism of motion query pair to model motion prediction as the joint optimization of the two tasks. Moreover, the interaction among agents is considered in the proposed method and collaborates to make the dense future prediction. The proposed method is demonstrated to be efficient on the large-scale Waymo Open Dataset. And an end-to-end variant of the proposed method is also provided for a broader study.

**Questions:**

I can understand the technical design of the proposed method while I have some concerns as mentioned in the "Weaknesses" part above. It would be helpful if the authors can address them with more clarification, with which I can more confidently assess the significance of this paper and adjust my rating.

**Limitations:**

Some technical design of the proposed method is explained at the end of the draft. I don't recognize more potential negative societal impact or limitations of this paper.

**Strengths And Weaknesses:**

Strengths:
1. The proposed method is well elaborated and the implementation details are necessarily provided to help understand the model design.
2. The performance of the proposed MTR on the Waymo Open Motion Dataset is good, advancing the SoTA under the setting further.
3. The paper is mostly well written that I can understand the motivation, high-level intuition, and model design quickly.

Weaknesses:
1. Some details are not clear, for example, the implementation of MTR-e2e is hard to follow for me given the limited illustration at L273.
2. Though it may not be necessary, it would be helpful to ablate the choice of query pair number for end-to-end version as well as there is a claim that "since 6 intention points are too sparse to well cover all potential future motions." may need some experiment backup.
3. Given the proposed method stresses on the design of query pair, it would be helpful for us to better understand the efficiency of this by making an ablation study of using only one type of query or both.

---

> ### Author Response · Authors · 2022-08-02
> **Response to Reviewer b34B**
>
> We sincerely thank the reviewer for providing thoughtful review and constructive suggestions. We answer the questions as follows.
>
> ****
>
> **Q1**: "The implementation of MTR-e2e is hard to follow given limited illustration."
>
> **A1**: We apologize for the confusion. Here we provide more implementation details of MTR-e2e by clarifying its two differences compared with the default setting MTR:
>
> * **Different numbers of motion query pairs**: MTR-e2e uses 6 motion query pairs (the required trajectory number in most benchmarks) so as to remove NMS post-processing, while MTR has 64 motion query pairs.
> * **Different training strategies for selecting positive motion query pair**: During the training process, we need to assign one motion query pair as the positive query, and its predicted trajectory is utilized for calculating the regression loss to cover the given ground-truth trajectory. MTR and MTR-e2e adopt different training strategies:
>     * MTR adopts a static assignment strategy that depends on 64 fixed intention points. As each motion query pair corresponds to one intention point, the motion query pair, whose intention point is closest to the endpoint of ground-truth trajectory, is selected as the positive query.
>     * MTR-e2e adopts a dynamic assignment strategy that depends on the predictions of 6 motion query pairs. As each motion query pair predicts a single future trajectory, we select the positive motion query pair by dynamically checking these 6 predictions whose endpoint is closest to the endpoint of the ground-truth trajectory. The reason for using such strategy is that the above static assignment strategy may assign a ground-truth trajectory to some faraway intention points and increase the burden of model optimization, since MTR-e2e only has 6 intention points that can not well cover all potential ground-truth future trajectories in a large region.
>
> Note that we ablate the effects of these two training strategies with different numbers of motion query pairs in Figure 4 and Section 4.3.
>
> ****
>
> **Q2**: "It would be helpful to ablate the choice of query pair number for end-to-end version."
>
> **A2**: Thanks for the valuable suggestion. As discussed in **A1**, the end-to-end version indicates using the dynamic assignment strategy to select positive motion query pair, which has been ablated in Figure 4 and Section 4.3. Specifically, the green lines in Figure 4 ablate the choice of query pair number under the dynamic assignment strategy,  where we can see that 6 motion query pairs achieve the best performance for MTR-e2e.
>
> ****
>
> **Q3**: "It would be helpful for us to better understand the efficiency of this by making an ablation study of using only one type of query or both."
>
> **A3**:
> Thanks for the constructive suggestion. We ablate the individual effects of the motion query pair by removing one of its two components. The results and discussions are listed below:
>
> | Intention Points | Static Intention Query  | Dynamic Searching Query  | Miss Rate (low) | mAP (high) |
> |:---------------:|:---------:|:---------:|:---------------:|:---------:|
> | Y               | Y         |  Y    |   **0.1668** | **0.3437** |
> | Y               | Y         |       |   0.1706 | 0.3284 |
> | Y               |           |  Y    |  0.1734  | 0.3379 |
> |                 |           |  Y    |  0.2150  | 0.2202 |
>
> * $1^{st}$ row: The default setting with complete motion query pair.
> * $2^{nd}$ row: Dynamic searching query in Eq.(7) is replaced with static intention query $Q_I$. The performance drops from 0.3437 to 0.3284 in terms of mAP.
> * $3^{rd}$ row: Static intention query in Eq.(6) is replaced with dynamic searching query $Q_S^j$, while intention points are still used to initialize dynamic searching query and select positive motion query pair for optimization. The performance slightly drops from 0.3437 to 0.3379 in terms of mAP.
> * $4^{th}$ row: The intention points are also removed. Hence, dynamic searching queries need to be set as learnable embeddings, and the dynamic assignment strategy is adopted to select positive motion query pair since we do not have any intention points. The performance drops dramatically from 0.3437 to 0.2202 in terms of mAP.
>
>
> The above experiments demonstrate that both two components of the motion query pair benefit the final performance, where static intention query is responsible for learning interaction among different motion modes (as in Eq.(6)) while dynamic searching query is able to probe trajectory-specific features for motion refinement (as in Eq.(7)).
> Notably, the intention points are critical for MTR framework since each motion query pair is bound with one specific intention point to better model multimodal motion prediction, and removing the intention points will greatly decrease the performance.
> Besides, all these settings have similar inference latency (33ms per scene as provided in L35 of supplementary material) since their basic frameworks remain unchanged.

---

> > ### Comment · Reviewer_b34B · 2022-08-07
> > **The added details and experiments are helpful**
> >
> > Thanks for the detailed explanation and additional results and clarification.
> >
> > The additional experiments to ablate the query setting is critical to demonstrate the efficiency of the proposed method given it is entangled with multiple components. I also read your responses to other reviewers and recognize the contribution made in this work. Most of my concerns are well addressed.
> >
> > I maintain my positive rating to this work. And will make more discussion in the next stage with other reviewers to calibrate my final rating.

---

> > > ### Author Response · Authors · 2022-08-08
> > > **Thanks for your positive feedback!**
> > >
> > > Thank you very much for acknowledging our additional experiments and providing positive feedback!
> > >
> > > Your constructive comments and suggestions are very helpful in improving our paper quality. Thanks!

---

### Meta-Review · Area_Chair_AdeF · 2022-08-25

**Recommendation:** Accept
**Confidence:** Certain

**Metareview:**

This paper proposes to model traffic vehicles using a transformer-based architecture for iteratively refining multimodal trajectory predictions. While the method is related to and builds upon several similar works in the area, it does also introduce some interesting new components such as the iterative refinement and the dynamic attention. Further, the strength of the experimental results from the combined system alone makes this paper important for researchers working in these areas: the method achieves the state of the art for trajectory prediction on two very widely used datasets (Waymo and Argoverse), compared to published leaderboards. All four reviewers unanimously agree that this paper is above the bar for acceptance, and I concur.

**Award:**

No

---

### Decision · Program_Chairs · 2022-09-14

Accept